

# Differences in the expression profiles of lncRNAs and mRNAs in partially injured anterior cruciate ligament and medial collateral ligament of rabbits

Huining Gu[1], Siyuan Chen[1], Mingzheng Zhang[2], Yu Wen[1] and Bin Li[2]

[1] Department of Histology and Embryology, College of Basic Medical Sciences, China Medical University, Shenyang, China
[2] Department of Joint Surgery and Sports Medicine, Shengjing Hospital, China Medical University, Shenyang, China

## ABSTRACT

Long noncoding RNAs (lncRNAs), as a novel regulatory factor, are considered to play a vital role in various biological processes and diseases. However, the overall expression profile and biological functions of lncRNAs in the partially injured anterior cruciate ligament (ACL) and medial collateral ligament (MCL) have not been clearly explored. Partially injured models of ACL and MCL were established in 3-month-old healthy male New Zealand white rabbits. Expression of lncRNAs and mRNAs in the ligament tissue was detected by high-throughput sequencing technology, and biological functions of differentially expressed RNAs were evaluated by Gene Ontology (GO) and Kyoto Encyclopedia of Genes and Genomes (KEGG) analysis. Validation of several differentially expressed RNAs was performed using quantitative real-time PCR (qRT-PCR). Protein-protein interaction (PPI) analysis and competitive endogenous RNA (ceRNA) prediction were used to identify interactions among hub genes and the interaction among lncRNAs, miRNAs, and mRNAs. The results showed that compared with the normal group, there were 267 mRNAs and 329 lncRNAs differentially expressed in ACL and 726 mRNAs and 609 lncRNAs in MCL in the injured group. Compared with MCL, 420 mRNAs and 470 lncRNAs were differentially expressed in ACL in the normal group; 162 mRNAs and 205 lncRNAs were differentially expressed in ACL in the injured group. Several important lncRNAs and genes were identified, namely, COL7A1, LIF, FGFR2, EPHA2, CSF1, MMP2, MMP9, SOX5, LOX, MSTRG.1737.1, MSTRG.26038.25, MSTRG.20209.5, MSTRG.22764.1, and MSTRG.18113.1, which are closely related to inflammatory response, tissue damage repair, cell proliferation, differentiation, migration, and apoptosis. Further study of the functions of these genes may help to better understand the specific molecular mechanisms underlying the occurrence of endogenous repair disorders in ACL, which may provide new ideas for further exploration of effective means to promote endogenous repair of ACL injury.

Corresponding authors
Yu Wen, ywen@cmu.edu.cn
Bin Li, surgeon_li@126.com

## INTRODUCTION

ACL is a connective tissue composed of bundles of collagen fibers, connects the femur and tibia, which prevents excessive anterior translation and internal rotation of the tibia with respect to the femur and provides mechanical stability to the knee joint (*Wang et al., 2020b*). MCL is the main stabilizing structure of the medial side of the knee, which resists valgus stress, provides static and dynamic stability, and assists in resisting rotational stress and anterior-posterior translation (*Guo et al., 2019*). Ninety percent of knee ligament injuries involve ACL or MCL, and present a commonly encountered problem in modern sports medicine (*Georgiev et al., 2018*). ACL injury is the leading cause of recurrent knee instability and may result in pain, limited range of movement, muscle weakness, biomechanical changes, and reduced physical activity level, while potentially affecting the function of other knee structures such as meniscal tears and articular cartilage degeneration, imposing a significant financial burden on the health care system (*Jia et al., 2017*; *Wang et al., 2020c*).

The self-healing response after ACL injury is very poor. Currently, the clinical treatment regarding ACL injury is mainly surgical repair or reconstruction. However, even after surgery, the ACL still cannot regain its normal biological functionality (*Xie et al., 2013a*; *Xie et al., 2014*; *Cai et al., 2017*). Recent evidence also indicates that postoperative inflammation may damage synovial stem cells and lead to an impaired joint environment, which inhibits tissue healing (*Wang et al., 2020c*), and even some patients may develop complications after surgery such as joint stiffness and osteoarthritis (OA) (*Xie et al., 2013a*; *Xie et al., 2014*; *Cai et al., 2017*). In contrast, injured MCL have a relatively better ability to self-heal and, in some cases, fully recover its function (*Furumatsu et al., 2010*; *Xie et al., 2013a*).

In recent years, scholars have proposed several possible explanations for the differences between the healing abilities of ACL and MCL after injury, including the different ultrastructural characteristics of connective tissue cells in ACL and MCL (*Lyon et al., 1991*), the differences in cellular properties and responses to growth factors between ACL and MCL (*Yoshida & Fujii, 1999*), MCL's improved capability to increase blood supply through increased angiogenesis and blood flow (*Bray, Leonard & Salo, 2003*); the different properties of ACL and MCL stem cells (*Zhang et al., 2011*); the differential expression of the LOX family, MMP-2, MMP-9, type I and V collagen and type III procollagen in ACL and MCL fibroblasts (*Xie et al., 2013a*; *Xie et al., 2013b*; *Georgiev et al., 2018*; *Georgiev et al., 2019*). However, the exact molecular mechanisms responsible for the endogenous repair difference of ACL and MCL are still unclear.

LncRNAs are a class of heterologous transcripts defined by sequence lengths exceeding 200 nucleotides and lacking any apparent protein-coding potential (*Kopp, 2019*). As a novel regulator, lncRNAs play key roles in biological processes such as cell proliferation, differentiation, apoptosis, autophagy, inflammation, and angiogenesis, as well as epigenetic, transcriptional, and post-transcriptional modifications (*Mousavi et al., 2013*; *Toiyama, Okugawa & Goel, 2014*), regulating the expression and function of protein-coding genes and participating in the pathogenesis of various types of diseases (*Gan et al., 2020*). In addition, the role of some lncRNAs in the inflammatory response, proliferation and apoptosis of certain human ligament cells has been demonstrated (*Wang et al., 2021*; *Zhou*

*et al., 2021*). Based on the above findings, we speculate that there may be differentially expressed lncRNAs involved in the regulation of genes related to ligament injury repair in ACL and MCL, resulting in the difference in healing ability between the two. However, the role of lncRNAs in knee ligament injury remains unclear, and there has been no exploration of the expression profiles, biological functions, or signaling pathways of lncRNAs in partially injured ACL and MCL. Therefore, we developed a partial injury model for rabbit ACL and MCL and examined the expression of lncRNAs and mRNAs in normal and partially injured ACL and MCL of rabbits by high-throughput sequencing. GO and KEGG enrichment analyses were performed to explore the potential biological functions and pathways of action of differentially expressed genes (DEGs). Additionally, PPI analysis and ceRNA co-expression network were used to identify hub genes and the interaction among lncRNAs, mRNAs, and miRNAs. We hope these results can provide a theoretical basis for exploring the possible molecular mechanism of endogenous repair disorder after ACL injury.

## MATERIALS & METHODS

### Animals and partially injured models of ACL and MCL

Three-month-old healthy male New Zealand white rabbits ($n = 6$) were purchased from Qingdao Kangda Biological Technology Co, Ltd. (Qingdao, China). Rabbits were bred at the Department of Experimental Animals of China Medical University (Shenyang, China) at $23 \pm 2$ °C with 12-hour light-dark cycles. The Medical Ethics Committee China Medical University approved the animal experiments on October 22, 2020 (approval number CMU2020310).

Animals were randomly divided into two groups with three rabbits in each group. Rabbits in the control group were bred under normal conditions and labeled N1, N2, and N3 respectively. The following treatment was done for the injured group: (1) Surgery to establish bilateral partially injured models of ACL and MCL: A longitudinal incision was made on the medial side of the knee joint of the rabbit, with an incision length of approximately 1.5 cm. The skin was incised layer by layer, and the subcutaneous tissue and joint capsule were bluntly separated to fully expose the ACL and MCL. ACL and MCL were laterally punctured by a 20ml needle at the proximal 1/3 of the femur and subsequently lacerated by a needle, resulting in blunt laceration of most (approximately2/3) of the ligament. The wounds were cleaned with physiological saline and the joint capsule and skin were sutured. (2) Postoperatively, ceftriaxone sodium 75 mg/kg was administered intramuscularly once daily for 3 days to combat the infection. Rabbits in the injured group were labeled as I1, I2, and I3. Rabbits in both groups were sacrificed to collect ligaments after deep anesthesia at 1 week after modeling. For all ligaments, snap-frozen in liquid nitrogen within 15min of surgery and stored at $-80$ °C. Two normal ACLs and MCLs and two injured ACLs and MCLs were randomly selected for RNA-Seq, and the rest of ligaments were used for qRT-PCR.

### RNA extraction, library construction, and RNA-Seq

Total RNA of each sample was extracted from the ligament tissue as per the instruction manual of the TRlzol Reagent (Life technologies, Carlsbad, CA, USA). RNA concentration was checked using Nanodrop2000 (Thermo Fisher Scientific, Waltham, MA, USA), and RNA integrity was checked using Agient2100 (Agilent Technologies Inc., Santa Clara, CA, MA, USA) and LabChip GX (PerkinElmer Inc., Waltham, MA, USA). RNA contamination was monitored on agarose gels. The cDNA libraries were constructed using the Ribo-off rRNA Depletion Kit after the samples passed the quality control, and then sequenced using Illumina NovaSeq 6000 platform.

### RNA-seq analysis

The high-quality reads were aligned to the Oryctolagus cuniculus reference genome (v2.Oryctolagus_cuniculus.v2.genome.fa) using HISAT2 software. StringTie (1.3.1) was used to calculate FPKMs of both lncRNAs and coding genes in each sample after completion of the comparison analysis. Differential expression analysis of two groups was performed using normalized count data which were derived using the DESeq2 package. Fold Change (FC) indicates the ratio of expression between two groups, and False Discovery Rate (FDR) is obtained by correcting the $p$-value for the significance of the difference ($p$-value) using Benjamini and Hochberg's approach. Genes with FDR < 0.01 and the absolute value of FC $\geq$ 2 were assigned as differentially expressed.

### Gene functional annotation

Gene function was annotated based on the following databases:Nr (NCBI non-redundant protein sequences); Pfam (Protein family); KOG/COG (Clusters of Orthologous Groups of proteins); Swiss-Prot (a manually annotated and reviewed protein sequence database); KEGG (Kyoto Encyclopedia of Genes and Genomes); GO (Gene Ontology). GO enrichment analysis was performed using the topGO R package, and Kobas software was used to test the statistical enrichment in the KEGG pathway.

### Construction of PPI network

The sequences of the DEGs were blast (blastx) to the genome of a related species (the protein-protein interaction of which exists in the STRING database: http://string-db.org/) to get the predicted PPI of these DEGs. Cytoscape software (version 3.8.2; https://cytoscape.org) was then used to build a PPI network and identify hub genes.

### Construction of lncRNA-miRNA-mRNA network

Competitive endogenous RNA (ceRNA) (*Salmena et al., 2011*) has gained attention in recent years as a new mode of transcriptional regulation. LncRNAs can be used as a miRNA sponge to compete with miRNAs and inhibit the regulation of miRNAs on the target gene, thus indirectly regulating gene expression (*Xiao et al., 2019*). Our study obtains candidate ceRNA relationship pairs by targeting relationships of miRNAs that satisfy the following conditions. (1) the number of identical miRNAs between ceRNAs should be greater than 5. (2) the $p$-value of the hypergeometric test is less than 0.01, and the corrected FDR value is less than 0.01. (3) The results of co-expression analysis are taken into account
to obtain a ceRNA co-expression network where both ceRNAs are co-expressed with each other. Based on the ceRNA network, the relationship pairs that differed in all three for each differential combination were extracted and visualized using Cytoscape software. The top 50 nodes in each group were identified using the cytoHubba plug-in in Cytoscape and Degree method mapped with Cytoscape.

## Quantitative real-time PCR

Total RNA of each sample was extracted from the ligament tissue according to the instruction manual of the TRlzol Reagent (Ambion, Massachusetts, USA). The PrimeScript II 1st Strand cDNA Synthesis Kit (Takara Bio inc., Kusatsu, Japan) and SYBR Premix ex Taq II (Takara) were used for reverse transcription and quantitative real-time PCR (qRT-PCR) assay. Polymerase chain reaction primer sequences are listed in Table S1 (Sangon Biotech Co., Ltd., Shanghai, China). β-Actin was applied as an internal control. For quantitative analysis of differential expression, data were processed by the $2^{-\Delta\Delta Ct}$ method and the expression of all lncRNAs and mRNAs were presented as relative fold changes to β-actin, which was repeated three times for each sample.

## Statistical analysis

All statistical data are expressed as the mean $\pm$ standard error of mean (SEM). Differences were analyzed by $t$-test using Graphpad Prism 8 statistical software. $P$-value < 0.05 was considered statistically significant.

## Data availability statement

All sequences are accessible *via* SRA using the accession PRJNA751479. Data is available at NCBI SRA, accession numbers: SRR15858594 to SRR15858601.

## RESULTS

### Statistical results of RNA sequencing data

The statistical results of RNA-seq data are shown in Table S2. A total of 31,217 genes were detected, and 10,903 new genes were discovered, 2,466 of which were functionally annotated. On the basis of the comparison results, gene expression analysis was carried out. Based on the expression of genes in different samples, 1,233 DEGs were identified, and functional annotation and enrichment analysis were performed. A total of 26,242 lncRNAs and 1,263 DELs were identified, and the obtained lncRNAs were divided into four groups with the following classification and percentage (Fig. S1): lincRNA (40%), antisense_lncRNA (8.8%), intronic_lncRNA (48.8%), and sense_ lncRNA (2.4%).

### DEGs screening, gene ontology, and Kyoto encyclopedia of genes and genomes pathway analysis
#### Differences between normal and injured groups

In total, 267 mRNAs were differentially expressed in the injured ACL compared with normal controls, of which 165 were up-regulated, and 102 were down-regulated (Figs. 1A and 1B). DEGs were enriched for several GO pathways such as cellular adhesion, positive regulation of angiogenesis, cellular response to tumor necrosis factor, elastic fiber assembly,
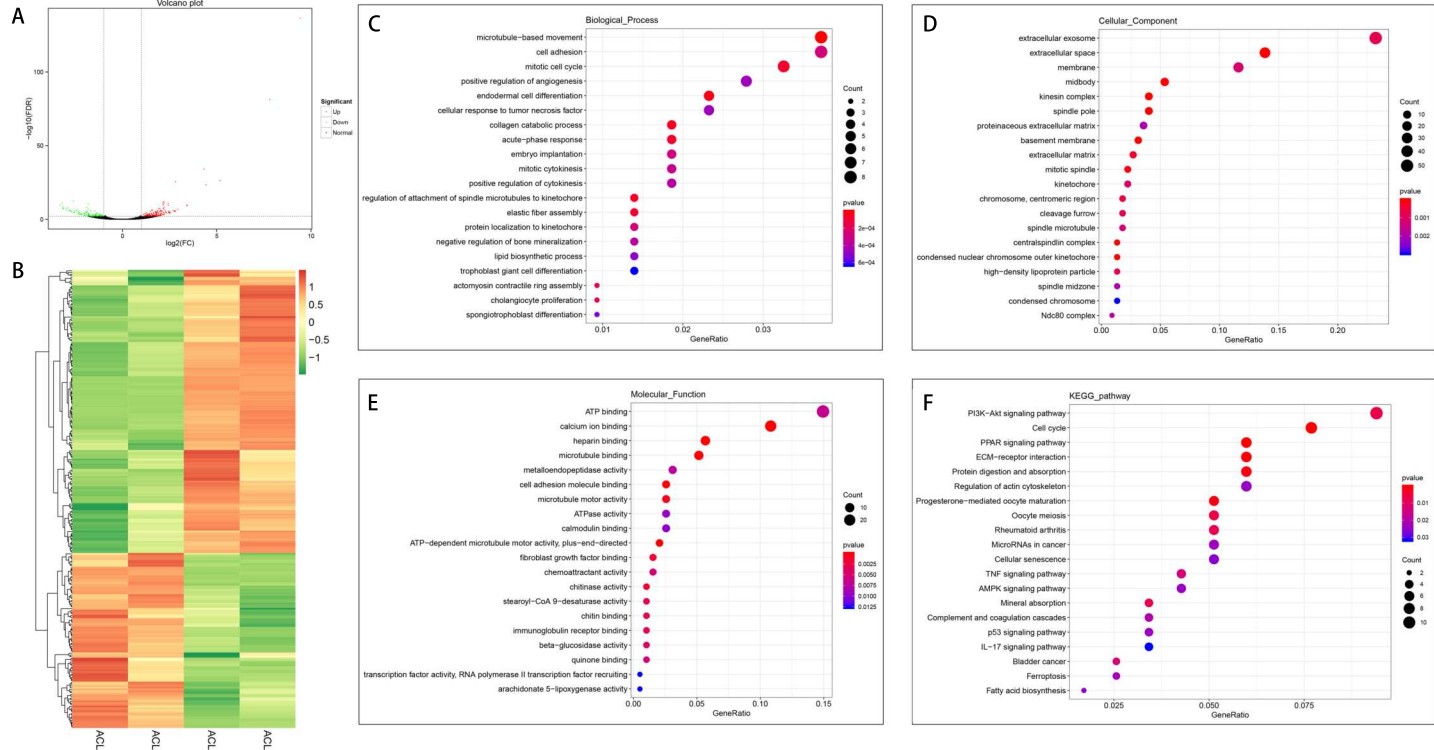

**Figure 1** **Comparative analysis of DEGs in the normal and injured groups of ACL.** (A) Volcano plot of the differentially expressed mRNAs. The red and green dots represent statistically significantly up-regulated and down-regulated mRNAs. (B) Hierarchical clustering shows a difference in mRNA expression profile between the two groups and homogeneity within groups. (C–E) Top 20 highest enriched GO terms for differentially expressed mRNAs. (F) Top 20 highest enriched KEGG pathways for differentially expressed mRNAs.

extracellular exosomes, proteinaceous extracellular matrix, extracellular matrix, cellular adhesion molecule binding, and fibroblast growth factor binding (Figs. 1C to 1E). The KEGG pathway enriched analysis results showed that DEGs were mainly enriched in the PI3K-Akt signaling pathway, cell cycle, PPAR signaling pathway, ECM-receptor interaction, Cellular senescence, AMPK signaling pathway, p53 signaling pathway, Ferroptosis, TNF signaling pathway, and the IL-17 signaling pathway, which were related to cell proliferation and apoptosis or inflammation (Fig. 1F).

A total of 726 mRNAs were differentially expressed in the injured MCL compared with normal controls, of which 372 were up-regulated, and 354 were down-regulated (Figs. 2A and 2B). DEGs were enriched for several GO pathways such as positive regulation of ERK1 and ERK2 cascade, inflammatory response, acute-phase response, blood vessel remodeling, phagocytosis, engulfment, positive regulation of phagocytosis, leucocyte migration involved in the inflammatory response, extracellular exosomes, cell surface, proteinaceous extracellular matrix, lysosome, extracellular matrix, cellular adhesion molecule binding, collagen binding, cytokine receptor activity, and collagen receptor activity (Figs. 2C to 2E). The KEGG pathway enriched analysis results showed that DEGs

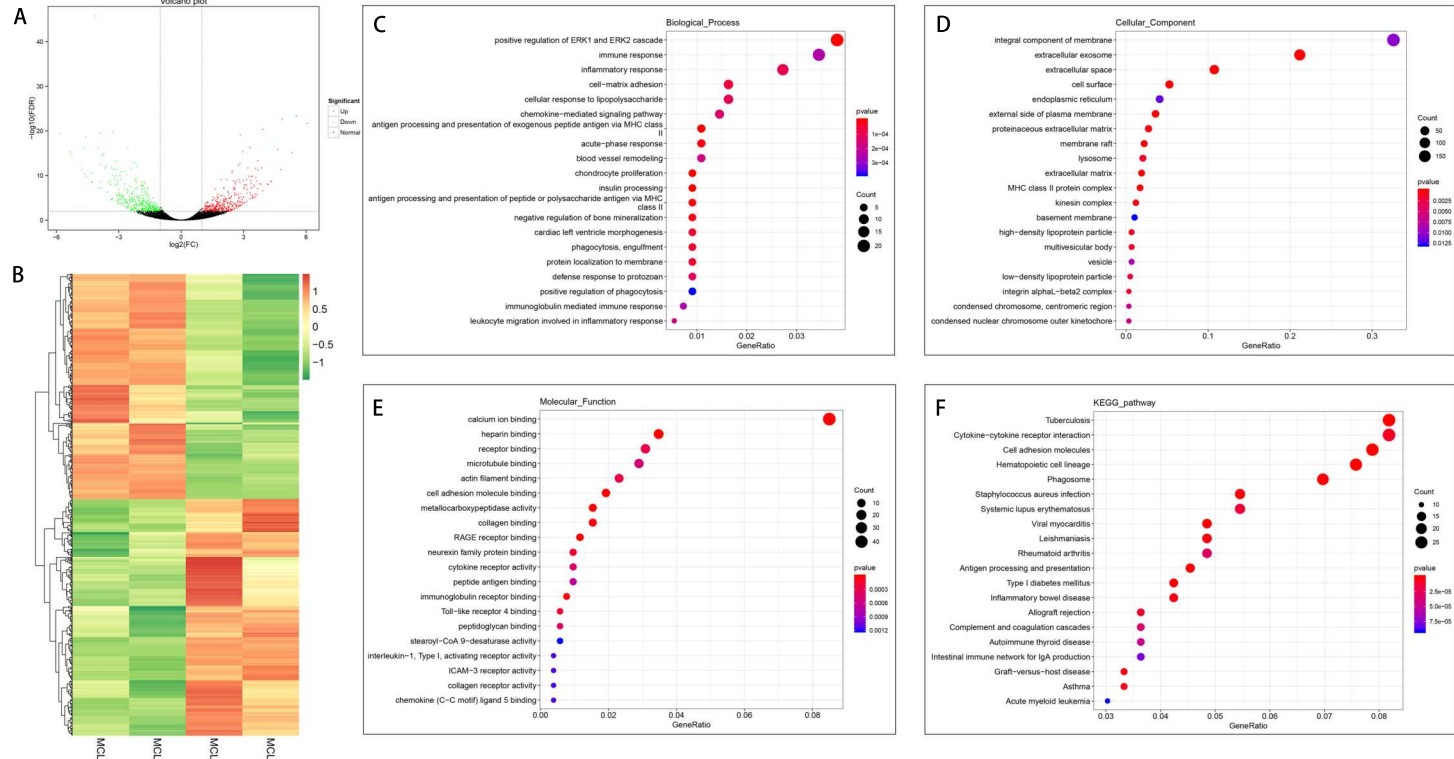

**Figure 2  Comparative analysis of DEGs in the normal and injured groups of MCL.** (A) Volcano plot of the differentially expressed mRNAs. The red and green dots represent statistically significantly up-regulated and down-regulated mRNAs. (B) Hierarchical clustering shows a difference in mRNA expression profile between the two groups and homogeneity within groups. (C–E) Top 20 highest enriched GO terms for differentially expressed mRNAs. (F) Top 20 highest enriched KEGG pathways for differentially expressed mRNAs.

were mainly enriched in Cytokine-cytokine receptor interaction, Cell adhesion molecules, and Phagosome, which were related to cell proliferation and apoptosis (Fig. 2F).

### Differences between the ACL and MCL groups

In the normal group, 420 mRNAs were differentially expressed in ACL compared to MCL, of which 178 were up-regulated and 242 were down-regulated (Figs. 3A and 3B). DEGs were enriched for several GO pathways such as extracellular matrix organization, cell adhesion, positive regulation of ERK1 and ERK2 cascade, blood vessel development, regulation of fibroblast growth factor receptor signaling pathway, extracellular exosome, proteinaceous extracellular matrix, cell surface, extracellular matrix, collagen type I trimer, collagen type IX trimer, collagen type II trimer, extracellular matrix structural constituent, growth factor activity, cell adhesion molecule binding, platelet-derived growth factor binding, and extracellular matrix binding (Figs. 3C to 3E). The KEGG pathway enriched analysis results showed that DEGs were mainly enriched in PI3K-Akt signaling pathway, Focal adhesion, ECM-receptor interaction, cell adhesion molecules, ErbB signaling pathway, FcγR-mediated phagocytosis, Leukocyte transendothelial migration, and TNF signaling

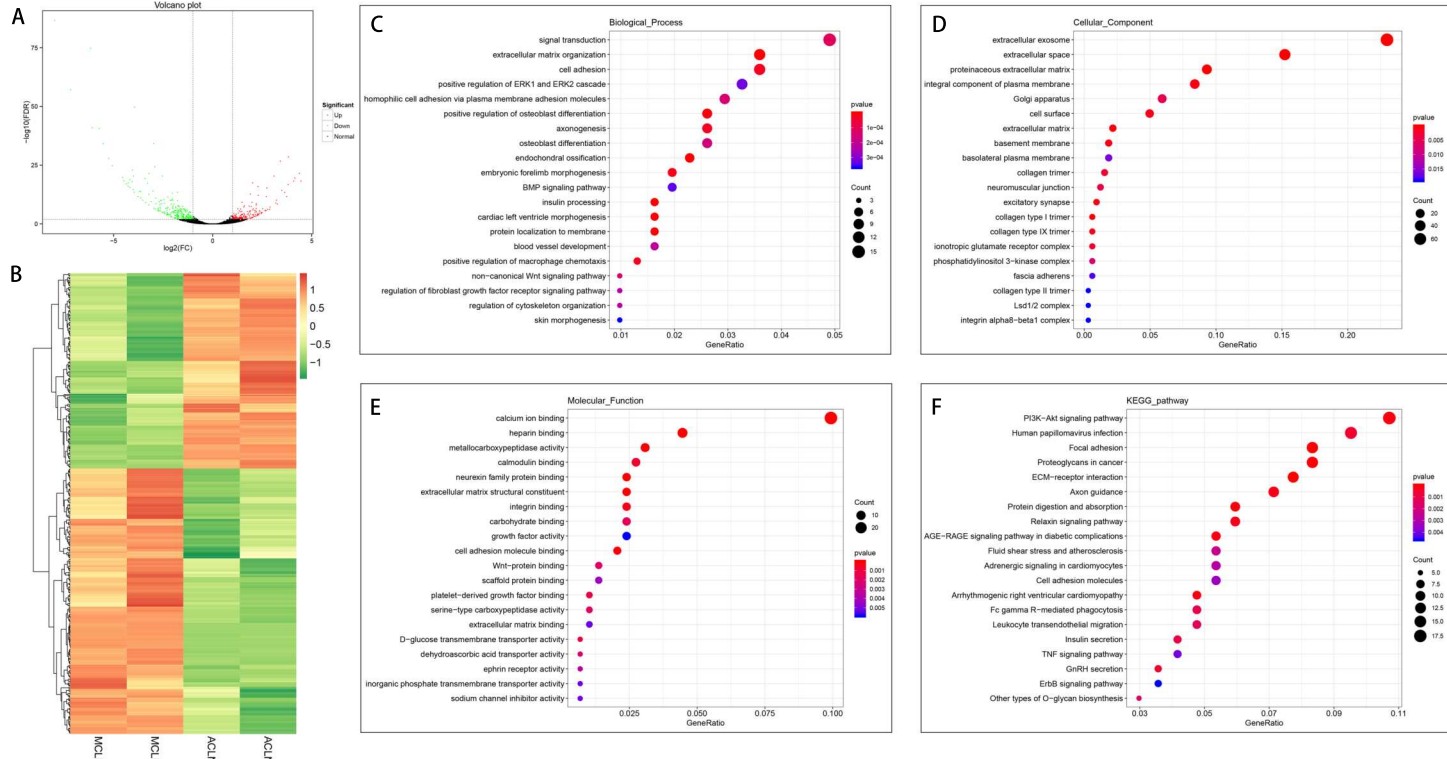

**Figure 3   Comparative analysis of DEGs in ACL and MCL of the normal group.** (A) Volcano plot of the differentially expressed mRNAs. The red and green dots represent statistically significantly up-regulated and down-regulated mRNAs. (B) Hierarchical clustering shows a difference in mRNA expression profile between the two groups and homogeneity within groups. (C–E) Top 20 highest enriched GO terms for differentially expressed mRNAs. (F) Top 20 highest enriched KEGG pathways for differentially expressed mRNAs.

pathway, which were related to cell proliferation, differentiation, migration apoptosis, and inflammation (Fig. 3F).

In the injured group, 162 mRNAs were differentially expressed in ACL compared with MCL, of which 82 were up-regulated and 80 were down-regulated (Figs. 4A and 4B). DEGs were enriched for several GO pathways such as elastic fiber assembly, positive regulation of mesenchymal stem cell differentiation, proteinaceous extracellular matrix, cell surface, elastic fiber, collagen type II trimer, and extracellular matrix structural constituent (Figs. 4C to 4E). The KEGG pathway enriched analysis results showed that DEGs were mainly enriched in ECM-receptor interaction, Focal adhesion, PI3K-Akt signaling pathway, and Cell adhesion molecules, which were related to cell proliferation and apoptosis (Fig. 4F).

## DELs screening, gene ontology, and Kyoto encyclopedia of genes and genomes pathway analysis
### Differences between normal and injured groups
In total, 329 lncRNAs were differentially expressed in the injured ACL compared with normal controls, of which 209 were up-regulated and 120 were down-regulated (Figs. S2A and S2B). Target genes of DELs were enriched for several GO pathways such as positive
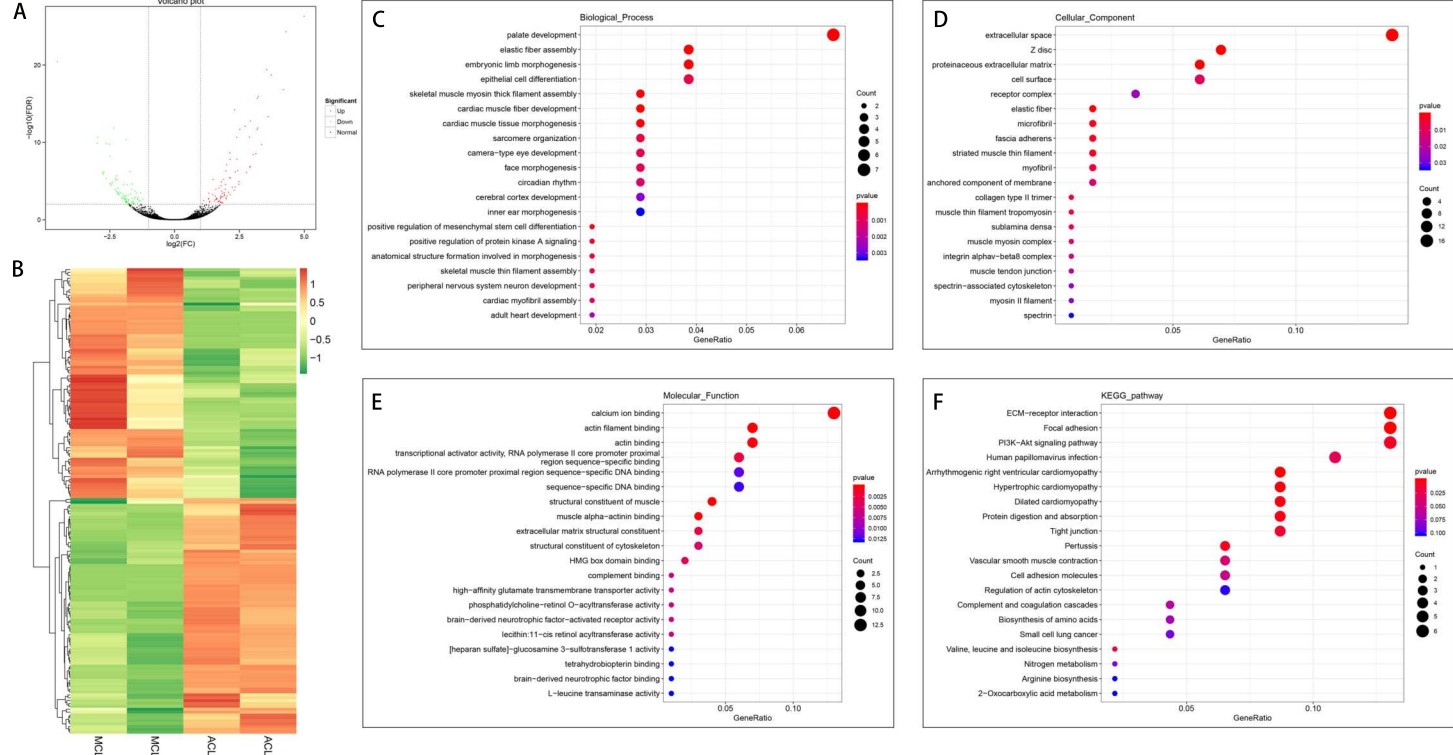

**Figure 4** **Comparative analysis of DEGs in ACL and MCL of the injured group.** (A) Volcano plot of the differentially expressed mRNAs. The red and green dots represent statistically significantly up-regulated and down-regulated mRNAs. (B) Hierarchical clustering shows a difference in mRNA expression profile between the two groups and homogeneity within groups. (C–E) Top 20 highest enriched GO terms for differentially expressed mRNAs. (F) Top 20 highest enriched KEGG pathways for differentially expressed mRNAs.

regulation of angiogenesis, positive regulation of inflammatory response, proteinaceous extracellular matrix, extracellular matrix, growth factor activity, chemokine activity, and cell adhesion molecule binding (Figs. S2C to S2E). The KEGG pathway enriched analysis results showed that target genes of DELs were mainly enriched in cytokine-cytokine receptor interaction, the cAMP signaling pathway, cGMP-PKG signaling pathway, chemokine signaling pathway, JAK-STAT signaling pathway, and the ErbB signaling pathway, which were associated with cell proliferation, differentiation, and migration apoptosis (Fig. S2F).

A total of 609 lncRNAs were differentially expressed in the injured MCL compared with normal controls, of which 263 were up-regulated, and 346 were down-regulated (Figs. S3A and S3B). Target genes of DELs were enriched for several GO pathways such as positive regulation of cell proliferation, cell division, positive regulation of angiogenesis, growth, platelet-derived growth factor receptor signaling pathway, extracellular exosomes, focal adhesion, proteinaceous extracellular matrix, growth factor activity, and cell adhesion molecule binding (Figs. S3C to S3E). The KEGG pathway enriched analysis results showed that target genes of DELs were mainly enriched in PI3K-Akt signaling pathway, cAMP signaling pathway, Focal adhesion, cell adhesion molecules, ECM-receptor interaction,

and ErbB signaling pathway, which were associated with cell proliferation, differentiation, and migration apoptosis (Fig. S3F).

### Differences between the ACL and MCL groups

In the normal group, 470 lncRNAs were differentially expressed in ACL compared to MCL, of which 166 were up-regulated and 304 were down-regulated (Figs. S4A and S4B). Target genes of DELs were enriched for several GO pathways such as positive regulation of ERK1 and ERK2 cascade, positive regulation of JNK cascade, regulation of cellular metabolic process, extracellular exosome, focal adhesion, proteinaceous extracellular matrix, stress fiber, growth factor activity, and cell adhesion molecule binding (Figs. S4C to S4E). The KEGG pathway enriched analysis results showed that target genes of DELs were mainly enriched in the PI3K-Akt signaling pathway, MAPK signaling pathway, Focal adhesion, cGMP-PKG signaling pathway, ECM-receptor interaction, and Ferroptosis, which were associated with cell proliferation, differentiation, and migration apoptosis (Fig. S4F).

In the injured group, 205 lncRNAs were differentially expressed in ACL compared to MCL, of which 116 were up-regulated and 89 were down-regulated (Figs. S5A and S5B). Target genes of DELs were enriched for several GO pathways such as acute-phase response, regulation of ERK1 and ERK2 cascade, and proteinaceous extracellular matrix (Figs. S5C to S5E. The KEGG pathway enriched analysis results showed that target genes of DELs were mainly enriched in the cAMP signaling pathway, focal adhesion, cGMP-PKG signaling pathway, ECM-receptor interaction, and the ErbB signaling pathway, which were associated with cell proliferation, differentiation, and migration apoptosis (Fig. S5F).

### Identifying candidate genes for PCR-validation

DEGs related to the altered pathway were further searched in the PubMed literature database. " Gene symbol "(gene name) was used as a query keyword and searched in the" title/abstract ". Three randomly selected up- and down-regulated genes associated with tissue damage repair were validated in each group. Finally, since very little information was available on DELs, three randomly selected DELs in each of the top five up-regulated and top five down-regulated DELs were chosen for further validation (Fig. 5). All the results of qRT-PCR are corresponded to RNA-seq data.

### Construction of lncRNA-miRNA-mRNA network

Based on the ceRNA regulatory network, the top 50 nodes in each group (Figs. 6A to 6D) were identified, including lncRNAs, miRNAs, and mRNAs(only 45 nodes in the normal group of ACL compared to the injured group). Each DEGs could be associated with one or more miRNAs and lncRNAs, which contained some genes related to the tissue damage repair process, such as COL7A1, LIF, FGD5, SLC20A1, and their associated lncRNAs and miRNAs were searched for, thus their networks were displayed with Cytoscape (Figs. 6E to 6G). The ceRNA network suggested that lncRNAs could not only participate in the regulation of biological processes independently but also act as miRNA sponges together to affect downstream gene expression and thus participate in the processes after ACL and MCL injury.
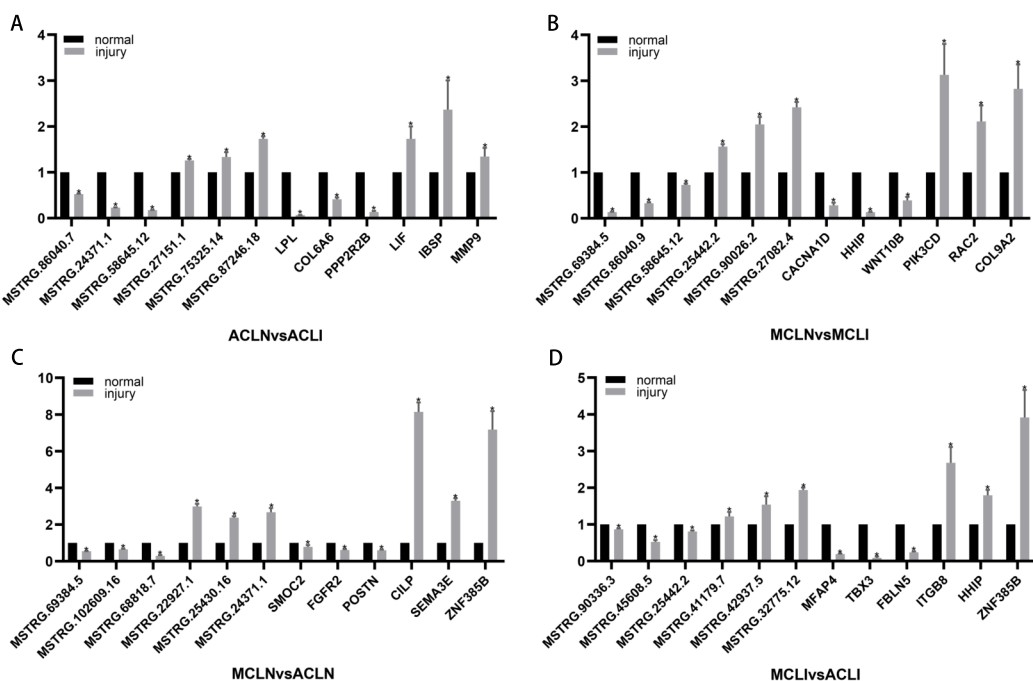

**Figure 5** **Validation of DEGs and DELs by qRT-PCR.** (A–B) Normal group *vs.* injured group (A) ACL, (B) MCL. (C–D) MCL *vs.* ACL. (C) normal group, (D) injured group. The *x*-axis represents the gene name, and the *Y*-axis represents the relative expression of DEGs and DELs. An asterisk (*) indicates significantly expressed DEGs and DELs.

## Construction of PPI network

We used PPI analysis and online STRING platform to detect the interaction effects between proteins in each group of DEGs, and then using cytoHubba plug-in and MCC method, we identified the top 30 ranked hub genes in each group (Figs. 7A to 7D), which contained several genes related to the tissue damage repair process, such as FGFR2, HBEGF, EPHA2, CSF1, MMP2, MMP9, SOX5, LOX.

## DISCUSSION

In this study, we simultaneously detected mRNAs and lncRNAs in normal and partially injured ACL and MCL of rabbits for the first time and provide a new gene resource for studying the role of lncRNAs in the development of partial injury in ACL and MCL. In addition, we established a ceRNA network and further analyzed DEGs using PPI to explore the specific molecular mechanism of endogenous repair disorder in ACL.

Ligaments are similar to tendons belonging to fibrous connective tissue, mainly composed of fibroblasts and extracellular matrix. The main component of the extracellular matrix of ligaments is neatly arranged type I collagen fibers, in addition to small amounts of other components such as elastin, decorin, dimeric glycans, and fibronectin (*Leong et al., 2020*). The repair of ligaments injuries is accomplished through a typical wound healing pathway that involves three main phases: inflammation, proliferation, and remodeling

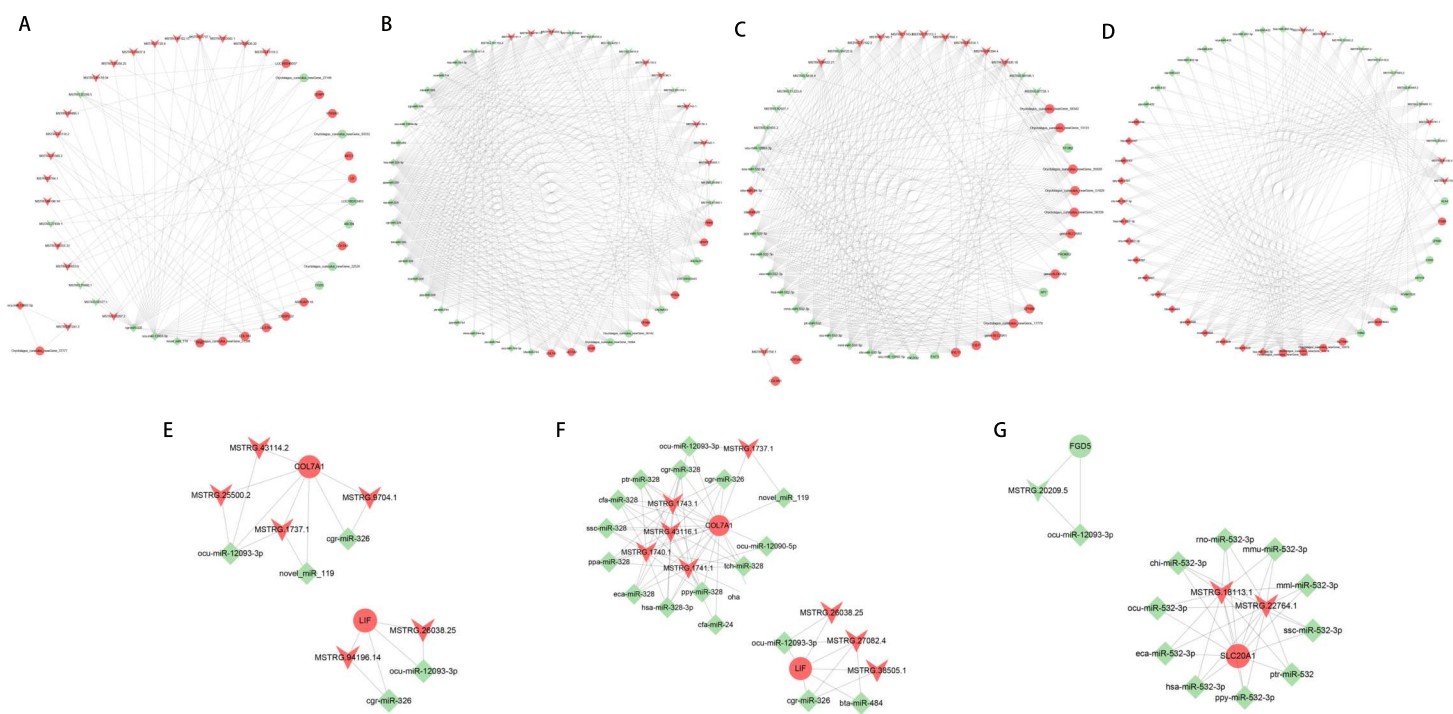

**Figure 6** **LncRNA-miRNA-mRNA ceRNA regulation network in each group(the top 50 nodes).** LncRNAs, miRNAs, and mRNAs are represented by arrows, diamonds, and circles, and up- and down-regulation are represented by red and green. (A–B) Normal group *vs.* injured group. (A) ACL, (B) MCL. (C–D) MCL *vs.* ACL. (C) Normal group, (D) injured group. (E–F) Regulatory network of COL7A1 and LIF. (E) ACL (F) MCL. (G) Regulatory network of FGD5, SLC20A1.

involving various cell types, cytokines, and extracellular matrix factors, and is closely related to inflammation, angiogenesis, the proliferation of fibroblasts, remodeling of the extracellular matrix, and the conversion of type III to type I collagen (*Cottrell et al., 2016*; *Lipman et al., 2018*). Our study showed that there were significant differences between ACL and MCL in terms of cellular composition, sensitivity to inflammatory responses, and cell proliferation and metabolic processes, which may be responsible for the differences in their healing capacity.

It is worth noting that pathways such as vascular remodeling, collagen-binding domains, and collagen receptor activity are specifically exhibited in the MCL after injury. Previous studies have shown that injured MCL has a better ability to increase blood supply relative to injured ACL (*Bray, Leonard & Salo, 2003*). Adequate blood supply is necessary for ligament healing, and alterations in peripheral vascular structures after acute injury can also promote hematoma for hemostasis purposes (*Lipman et al., 2018*). The enrichment of vascular remodeling in the injured MCL in GO analysis may account for its ability to have a better blood supply, which may have contributed to the different healing abilities of the ACL and MCL after injury. The main extracellular matrix component of ligaments is collagen, and the process of remodeling in the late stages of ligament injury is closely related to the remodeling of the extracellular matrix and the conversion of type III collagen to type I collagen (*Cottrell et al., 2016*; *Lipman et al., 2018*). The enrichment of collagen-binding

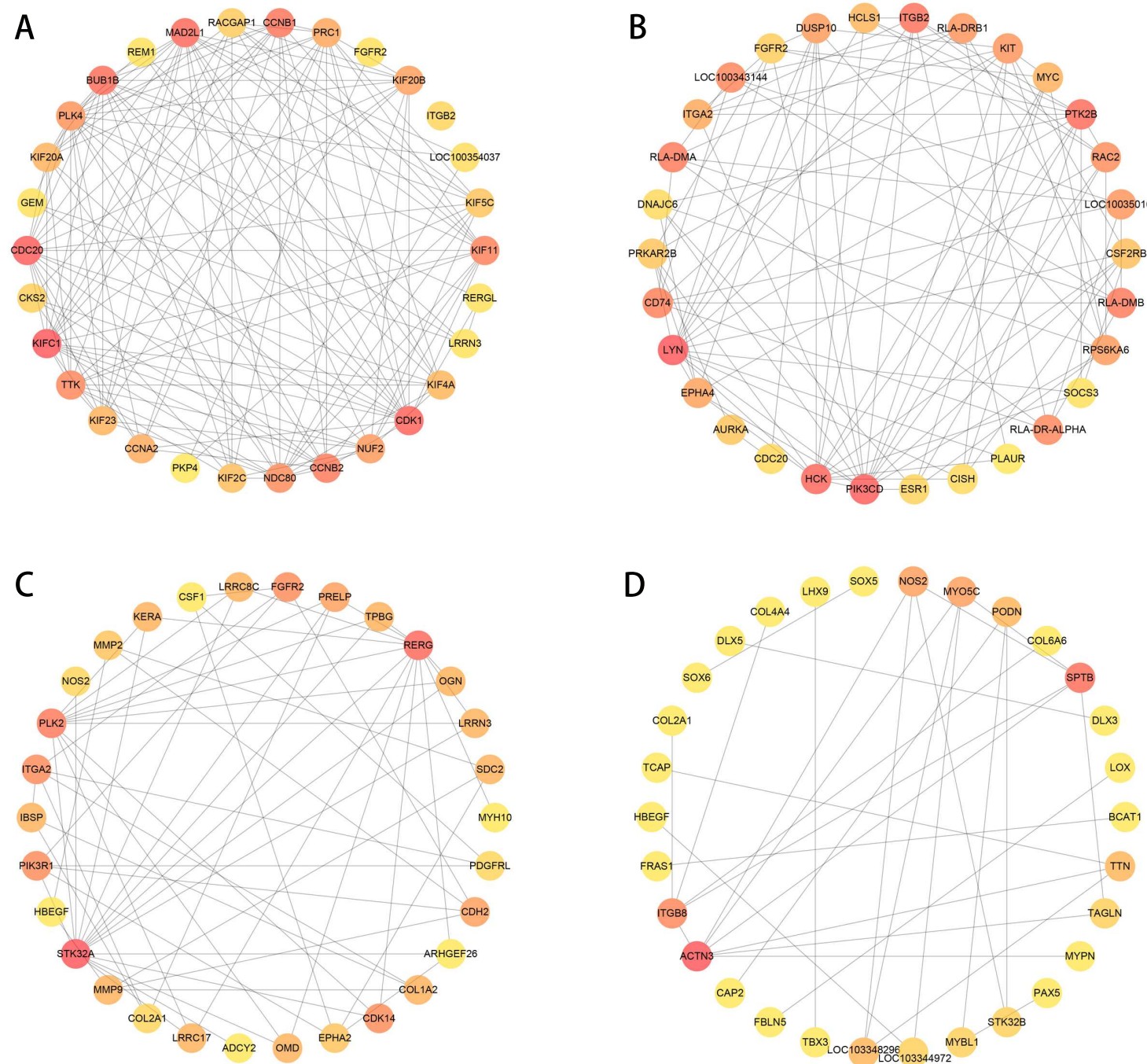

**Figure 7** **PPI network analysis in each group of DEGs (top 30 hub genes).** The higher score has a red color, and the lower score has a yellow color. (A–B) normal group *vs.* injured group. (A) ACL, (B) MCL. (C–D) MCL *vs.* ACL. (C) normal group, (D) injured group.

domains and collagen receptor activity in the injured MCL in GO analysis may be more favorable for ligament repair in the late stages of injury. In addition, KEGG analysis showed that DEGs and the target genes of DELs of injured ACL appeared enriched in the IL-17 signaling pathway, cellular senescence, p53 signaling pathway, ferroptosis, and the JAK-STAT signaling pathway. As a pro-inflammatory cytokine, IL-17 exerts its effects by promoting the expression of matrix metalloproteinases and proinflammatory genes, which can result in neutrophil infiltration and inflammation in tissue (*Iwakura et al., 2011*). Ferroptosis is an iron-dependent, novel form of programmed cell death accompanied by massive iron accumulation and lipid peroxidation (*Li et al., 2020*). P53 signaling pathway regulates a variety of cellular processes, including apoptosis, growth inhibition, cell cycle progression inhibition, cellular senescence after stress, autophagy, and apoptosis (*Jeffries & Krupenko, 2018*). The JAK-STAT signaling pathway is a signal transduction pathway stimulated by cytokines involved in many critical biological processes such as cell proliferation, differentiation, apoptosis, and immune regulation (*Xin et al., 2020*). Studies have shown that regulated inflammation largely facilitates tissue repair, while excessive or persistent inflammation may lead to poor tissue repair outcomes (*Thomopoulos et al., 2015*). The enrichment of the above pathways seen in the injured ACL may indicate that there may be a more severe inflammatory response after ACL injury and that more cell death pathways are activated, resulting in a poorer healing capacity.

Subsequently, we established a ceRNA network and further analyzed it, obtaining some important nodes such as COL7A1, LIF, FGD5, SLC20A1, and the regulatory network associated with them. COL7A1 plays a key role in physiological wound healing, supporting dermal fibroblast migration and regulating their cytokine production in granulation tissue, contributing to skin wound healing, loss of COL7A1 harms both the process and outcome of skin healing (*Nyström et al., 2013*). LIF is a glycoprotein in the interleukin-6 cytokine family, which has pleiotropic actions throughout the body, affecting stem cell self-renewal, cell proliferation, differentiation and survival (*Metcalf, 2003*). Recent studies have shown that in human myogenic cells, LIF can increase the number of myogenic cells by increasing mitosis and decreasing apoptosis (*Broholm et al., 2012*) and that LIF has neurodegenerative and protective functions in perinatal hypoxic-ischemic brain injury (*Lin et al., 2020*) and has some pro-angiogenic potential (*Santos et al., 2020*). Our results showed a trend of upregulation of both COL7A1 and LIF in ACL and MCL after injury, perhaps with positive effects on healing after injury in both, but with differences in their degree of change and regulatory networks. In addition, FGD5 can regulate the pro-angiogenic effect of VEGF in vascular endothelial cells (*Kurogane et al., 2012*), which showed a trend of down-regulation in ACL after injury, which might also have an impact on the recovery of hemopoietic capacity after ACL injury. SLC20A1 may promote the production of pro-inflammatory and chemotactic mediators and ROS (*Koumakis et al., 2019*), and in a comparison of normal group ACL with MCL, SLC20A1 was found to be significantly upregulated in ACL, and such differences may make ACL more prone to more inflammatory responses. Meanwhile, we have obtained several important lncRNAs associated with the regulation of these genes on the basis of the ceRNA network, including MSTRG.1737.1, MSTRG.26038.25, MSTRG.20209.5, MSTRG.22764.1, and MSTRG.18113.1. These lncRNAs might serve as

non-coding gene candidates for further study of the pathogenesis of endogenous repair disorders in ACL. Elucidation of their interactions with the corresponding candidate genes would help to better understand the molecular mechanisms by which lncRNAs regulate the occurrence of endogenous repair disorders in ACL.

We additionally used PPI analysis to predict the interaction effects between the proteins of each group of DEGs, and then obtained some hub genes such as FGFR2, EPHA2, CSF1, MMP2, MMP9, SOX5, LOX. FGFR2b is involved in maintaining the stability of the internal skin environment and wound healing (*Katoh, 2009*), and previous studies on ligament healing have also shown that FGFR2 plays a role in the proliferation of injured ligament fibroblasts and endothelial cells (*Rösler, 1979*). EPHA2 has been reported as an important factor in regulating cell permeability and tight junctions in brain endothelial cells, and the inactivation of EPHA2 promotes tight junction formation and impairs angiogenesis in brain endothelial cells (*Zhou et al., 2011*). EPHA2/Ephrin-A1 signaling complexes restrict the migration of corneal epithelial cells (*Kaplan et al., 2012*). Recent studies have shown that the EPHA2 signaling pathway is involved in the permeability and inflammatory response to lipopolysaccharide-induced lung injury, and the application of EPHA2 monoclonal antibody to inhibit the expression of EPHA2 can reduce lung injury (*Hong et al., 2016*). CSF1 influences the environment of renal epithelial cell growth, proliferation, and differentiation, and promotes matrix remodeling and cell replacement during inflammation, contributing to kidney growth and endogenous repair following injury (*Alikhan et al., 2011*). SOX5 acts as a transcription factor involved in regulating embryonic development and determining cell fate (*Chen et al., 2018*), as well as in regulating differentiation, proliferation, and neuronal development. Recent studies suggest the neuroprotective effects of SOX5 against ischemic stroke by regulating VEGF/PI3K/AKT pathway (*Zhang et al., 2021*). MMP2 and MMP9 degrade extracellular matrix (ECM) proteins, such as type IV and V collagen, elastin, and vitreous junction proteins, and play key roles in connective tissue remodeling, demyelination and remyelination after injury, inflammation and glial cell reactivity, cell genesis and migration, and angiogenesis (*Verslegers et al., 2013*; *Xie et al., 2018*) Recent studies have found that MMP9 mediates the changes in collagen fibril organization that occur in acute and chronic tissue injury and has a regulatory effect on collagen structure after trauma (*LeBert et al., 2015*). LOX is a copper-dependent amine oxidase that plays a key role in matrix synthesis by catalyzing the formation of crosslinks between collagen and elastin fibers (*Lucero & Kagan, 2006*). In recent years, studies on the differential healing ability of injured ACL and MCL have also suggested possible explanations for the differential expression of MMPs and LOX on several occasions (*Xie et al., 2013a*; *Georgiev et al., 2018*). Several previous studies have demonstrated that imbalance in ECM biosynthesis and degradation caused by altered LOX and MMP expression may be responsible for the defective ACL repair (*Beye et al., 2008*; *Xie et al., 2013b*; *Zhang et al., 2017*; *Wang et al., 2020a*) which correspond to our analysis of the RNA-seq data. Further exploration of the role of these genes after ACL injury may be more beneficial to our understanding and search for specific molecular mechanisms of endogenous repair disorders in ACL.

Our study has some limitations. Experimental validation of the putative ACL endogenous repair disorder-related genes and lncRNAs warrants further investigation. The small reproducibility of the samples may result in incorrect screening of specific lncRNAs and genes, and further studies will verify their reproducibility and reliability. Similarly, efforts should be made to determine the specific roles of other DEGs, and DELs identified in this study and to place them in the larger context of the progression of endogenous repair disorders in ACL.

## CONCLUSIONS

In conclusion, our study provides the first preliminary analysis of lncRNAs and mRNAs differentially expressed in normal and injured groups of ACL and MCL and predicts several important lncRNAs and genes, which may serve as gene candidates for further study of the pathogenesis of endogenous repair disorders in ACL. Further study of the functions of these genes and their interactions with lncRNAs may help to better understand the specific molecular mechanisms underlying the occurrence of endogenous repair disorders in ACL, which may provide new ideas for further exploration of effective means to promote endogenous repair of ACL injury.

### Funding

This work was supported by the Shenyang Bureau of Science and Technology under Grant number 20-205-4-078. The funders had no role in study design, data collection and analysis, decision to publish, or preparation of the manuscript.

### Grant Disclosures

The following grant information was disclosed by the authors:
Shenyang Bureau of Science and Technology: 20-205-4-078.

### Competing Interests

The authors declare there are no competing interests.

### Author Contributions

- Huining Gu conceived and designed the experiments, performed the experiments, analyzed the data, prepared figures and/or tables, authored or reviewed drafts of the paper, and approved the final draft.
- Siyuan Chen and Mingzheng Zhang performed the experiments, prepared figures and/or tables, and approved the final draft.
- Yu Wen and Bin Li conceived and designed the experiments, authored or reviewed drafts of the paper, and approved the final draft.

### Animal Ethics

The following information was supplied relating to ethical approvals (i.e., approving body and any reference numbers):

All animal experiments were conducted according to legal regulations in China and were carried out with permission and under the regulation of the Medical Ethics Committee China Medical University.

## DNA Deposition

The following information was supplied regarding the deposition of DNA sequences:

All sequences are available at SRA: PRJNA751479.

## Data Availability

The data is available at NCBI SRA: SRR15858594 to SRR15858601.

## Supplemental Information

Supplemental information for this article can be found online at http://dx.doi.org/10.7717/peerj.12781#supplemental-information.

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
