# Peer review of "Differences in the expression profiles of lncRNAs and mRNAs in partially injured anterior cruciate ligament and medial collateral ligament of rabbits"

_PeerJ, doi:10.7717/peerj.12781_

## Round 0.1 · original submission · Major Revisions

Thank you for this interesting data. According to the comments of reviewers, many points need revision.

Reviewer 1 ·

Basic reporting

The paper was easy to follow and comprehend. I found no issues with use of English language and grammar.

All the figures used in the paper are high quality and annotated with appropriate legends and titles.

Experimental design

The authors have stated all the methods with clarity along with specific versions of Bioinformatics tools used. I am impressed by the extensive work that was done on RNA-seq analysis, Gene functional annotation and PPI network construction. The authors also very clearly state all the threshold values and cut-offs used for each of the bioinformatics analysis. I have also thoroughly checked that the raw data has been deposited in SRA.

Validity of the findings

The findings are compelling and backed by extensive application of bioinformatics tools and necessary statistical to show the involvement of specific lncRNAs in the pathogenesis of endogenous repair disorders in ACL.

Reviewer 2 ·

Basic reporting

1. “Introduction”: Although orthopedic researchers are very familiar with ACL and MCL, it would be nice if there is some background information about ACL and MCL, and their differences in this part.
2. The label in the Figure2,3,4,5,6,7,8 should be only A, B C, D, instead of Figure Number A.
3. The structure could be more concise. For example, “Line 175-180 would be better in the “Methods” section.
4. There are some grammatical errors. English revision is necessary.

Experimental design

1. The description of ACL, MCL, and the normal group is unclear and confusing. According to the “methods” section, 3 rabbits were normal, and 3 rabbits had ACL and MCL injuries. How is the injury? Are MCL and ACL injuries given on the same rabbit? Then according to the results part, and Figure2, Figure3, it seems that there are 2 animals in each group. What is the difference between ACLN1, ACL N2; MCL N1, MCL N2; and the N1, N2, N3? Please clearly describe the definition of the normal controls and the injured groups.
2. The authors selected some random genes to verify their RNA-seq. It would be more meaningful to verify the most important genes related to the altered pathway, which would also make up for the limitations.

Validity of the findings

There are reports on the difference in gene expression between ACL and MCL injuries (Jasmine A Beye et al. Am J Sports Med. 2008. PMID: 18448582.) and molecular changes of ACL obtained from injury animal models or patients by gene sequence methods. Authors should include these citations in their papers to have a comparison.

Reviewer 3 ·

Basic reporting

The English expression is clear and could be improved with more of background information for why this is important and how this correlate to clinical findings/problems.

There are no hypothesis in the manuscript.

Experimental design

In Gu et al. manuscript “Differences in the expression profiles of lncRNAs and mRNAs in partially injured…”, authors introduced anterior cruciate ligament (ACL) and medical collateral ligament (MCL) to rabbits and performed RNA-seq for ligament tissues. By comparing their lncRNA expression profiles, significantly altered lncRNAs were analyzed with pathways enriched, and partially confirmed with RT-PCR. While seems like there are differences, the manuscript could be further improved if answering following questions:
1. How ACL and MCL in rabbits related to clinical human disease? Is there any supportive evidence showing the model applied in the paper could mimic clinical ACL and MCL?
2. What is the central hypothesis of the manuscript and why it is important?
3. Most of figures in the manuscript are too small to read, it could be improved if the figures could be reorganized with the key findings in main figures and others in supplement.

Validity of the findings

4. Are RT-PCR confirmed expression correspond to RNA-seq data? What about most differentially expressed gene but not yet discussed as a title in literature?
5. Is there any protein level confirmation or clinical samples could confirm the differential expressed lncRNAs/pathway related genes?

Reviewer 4 ·

Basic reporting

This manuscript addressed the expression profiles of lncRNAs and mRNAs in partially injured ACL and MCL of rabbits at 1week after modeling. The authors mentioned the importance of sequencing of the lncRNAs, miRNAs, and mRNAs in intact and partially injured ACL and MCL. The point of this manuscript is newest and prevalent. However, the results and discussion sections have not been configured.
This manuscript has many concerning points to discuss the results. Before thoroughly evaluating, the authors should be improving some issues of methodology and re-consider the section of discussion of this study.

Experimental design

Line 71-74
The authors should clearly explain why to analyze the expression of lncRNA and protein-protein interaction (PPI) analysis in this study. This study design confuses the reader. Please clearly state the purpose of this analysis.

Line 113
Why do authors use FPRMs? FPKM does not respect invariance property and cannot be an accurate measure of relative molar RNA concentration (See: Wagner et al. "Measurement of mRNA abundance using RNA-seq data: RPKM measure is inconsistent among samples" Theory in Bioscience. Vol.131, pp281-285, 2012.). Authors should re-calculate using Transcripts Per Million (TPM).

Validity of the findings

Line 305-
I understand that this research consisted of many results. However, the discussion section replaced the enumeration of the results. There is not enough interpretation of the results. Authors should be reconstructing and brushing up

Line 330-335
Too long sentence. Please re-consider.
Many sentences in the discussion section are too long. Authors should be carefully confirming the whole manuscript.

Line 354-
In this study, the authors compared intact ACL with MCL in the normal group. In this case, the authors should mention the baseline difference of ACL and MCL genes.

Additional comments

Through the discussion section,
This manuscript lacks the assessment of results in the discussion section. Many of the results showed, however, the author's arguments are not clear. Reduce the listed gene names and reconstruct the arguing point.

---

## Round 0.2 · accepted · Accept

Congratulation on your manuscript.

Reviewer 2 ·

Basic reporting

The authors made some changes according to the modification comments, which is helpful to the improvement of the paper.
However, there are some problems to be further improved as well:
1. The quality and clarity of the figure.
2. The title of each section in “Results” should be clear and represent the key points, not a description of the used methods.

Experimental design

no comment

Validity of the findings

no comment

Reviewer 3 ·

Basic reporting

Authors had addressed all my concerns and questions.

Experimental design

Authors had addressed all my concerns and questions.

Validity of the findings

Authors had addressed all my concerns and questions.

Reviewer 4 ·

Basic reporting

The authors have very thoroughly addressed all of my previous concerns.

Experimental design

The authors have very thoroughly addressed all of my previous concerns.

Validity of the findings

The authors have very thoroughly addressed all of my previous concerns.

Additional comments

There are no comments anymore.